# Comparative SIR/SEIR modeling of the Antonine Plague in Rome

**Anestis Karasaridis**[1,2]*, **Aleš Chalupa**[1,2]

**1** Department for the Study of Religions, Faculty of Arts, Masaryk University, Brno, Czech Republic, **2** Centre for the Digital Research of Religion (CEDRR), Masaryk University, Brno, Czech Republic

* anestis.karasaridis@mail.muni.cz

**Data Availability Statement:** All code used to complete the simulations and analyses is publically available on GitHub (https://github.com/ane57is/

## Abstract

Some scholars consider the Antonine Plague to have been a major disease outbreak in the 2nd century CE that caused a significant decline in the population of the Roman Empire. Although there is currently no molecular evidence of the specific pathogen, literary evidence indicates the parameters of the disease that it caused and how significant the impact on Roman society was. One way to advance the current discussion concerning the Antonine Plague's impact on the Roman Empire's population is to examine the currently available sources and comparatively model the spread of different pathogens in a specific location with known demographic data for the relevant period. To accomplish this, we developed a series of dynamic ordinary differential equation models of the spread of disease in Rome between 165 and 189 CE for several pathogens. We found that daily disease deaths in the final years of the pandemic were inconsistent with estimates reported in primary sources, suggesting that either (a) the impact of the Antonine Plague may have been exaggerated in the descriptions of ancient authors, or (b) the daily deaths in ca. 189 CE were caused by a different disease event than the Antonine Plague, or (c) seasonality might have been a significant factor changing the intensity of disease spread, with the population more severely affected during the winter months. Although none of the pathogens we analyzed emerged as the likely causative agent of the Antonine Plague, the models show that the overall mortality rate would have increased maximally by 7%. This result contradicts the mortality rate accepted by historians who defend the thesis of the significant impact of this epidemic on the demography of the Roman Empire.

## Introduction

William H. McNeill, in his book *Plagues and Peoples*, suggested that microparasitism, i.e., the spread and impact of infectious diseases, was a significant factor in the "[r]eligious and cultural history of the [Roman E]mpire as well as [in] its social and political development" [1]. Some scholars consider the Antonine Plague to have been a major disease outbreak in the 2nd century CE that caused significant damage to the population of the Roman Empire. Despite the current lack of molecular evidence for the specific pathogen, literary evidence provides some information about the symptoms of the disease that it caused and how significant the impact

ap_epidemiology_2021) and Zenodo (https://zenodo.org/records/13857741).

**Funding:** This work was supported by the project "Interní grantová agentura Masarykovy univerzity", CZ.02.2.69/0.0/0.0/19_073/0016943, was part of the project "The Antonine Plague in Rome: Comparative Modeling of Different Pathogens", MUNI/IGA/1335/2020, and was supported by the European Regional Development Fund project "Beyond Security: Role of Conflict in Resilience-Building" (reg. no.: CZ.02.01.01/00/22_008/0004595). The funders had no role in study design, data collection and analysis, decision to publish, or preparation of the manuscript.

**Competing interests:** The authors have declared that no competing interests exist.

on Roman society was. Although the current debate favors smallpox as the causative agent (pathogen) of the Antonine Plague, other pathogens (e.g., measles) are also being considered.

Scholars have interpreted the extent and impact of the Antonine Plague in different ways. On the basis of sources from Egypt and Italy, Richard Phare Duncan-Jones argued that the Antonine Plague caused a significant economic decline throughout the Roman Empire after 165 CE [2]. Walter Scheidel supported Duncan-Jones's position, emphasizing a link between the increase in mortality and the transformation of rents and wages [3]. Because of their interpretations of the Antonine Plague as a severe blow to the Roman Empire, Duncan-Jones and Scheidel's position is sometimes considered maximalist [4].

In contrast to these views, some scholars take a minimalist position. James Frank Gilliam argued that the demographic impact of the pandemic has been overestimated (both in historical sources and by other researchers) [5]. Similarly, using the same sources and data but interpreting them differently, Roger S. Bagnall [6] criticized the work of Duncan-Jones [2] and Scheidel [3]. According to Bagnall, the pandemic did not have a significant impact on the subsequent transformation of the population of the Roman Empire. Therefore, alternative hypotheses about the events that might have triggered the demographic change in the Roman Empire should be considered and tested.

The diagnosis of the Antonine Plague has been debated for decades, with smallpox, bubonic plague, and measles being proposed [7–9]. Due to the current lack of molecular evidence for the cause of the Antonine Plague, such diagnoses are constructed retrospectively on the basis of literary evidence. The authors contemporary with the outbreak are Galen, Marcus Aurelius, Lucian, Aristides, Cassius Dio, and Philostratus. In addition, several other authors wrote retrospectively about the Antonine Plague in the 4th and 5th centuries CE [5]. Based on such accounts, the reconstruction of the possible disease that spread throughout the Roman Empire during the Antonine Plague is inevitably speculative. One line of research that might help to overcome this lack of knowledge is the comparative modeling of the spread of different pathogens in a specific location with known demographic data for the relevant period (such as the city of Rome between 165 and 189 CE).

To the best of our knowledge, the only Empire-wide computational analyses have been carried out by Yan Zelener (without using the most recent population estimates) [10]. An epidemic network modeling study has been conducted by Sean F. Everton and Robert Schroeder, but it does not model the specific situation of the city of Rome [11]. Since our study focuses only on the spread of the Antonine Plague in Rome, the results may be affected by fewer confounding variables than in the case of Empire-wide modeling. Note that Zelener [10] and Everton and Schroeder [11] used models based on the current consensus regarding smallpox as the cause of the Antonine Plague; the specific pathogen may have been different. Fortunately, Lauren A. White and Lee Mordechai's recent article laid the groundwork for a methodological framework for modeling different plague dynamics in the case of the Justinianic Plague in Constantinople [12], which can easily be adapted to model the dynamics of the Antonine Plague in Rome. Recent epidemic modeling studies using SIR/SEIR frameworks have demonstrated the utility of these models in understanding disease dynamics (e.g., COVID-19), but our approach adapts these models to a historical context to explore the specific case of the Antonine Plague in Rome [13–19]. As there is currently no molecular evidence for the causative agent of the Antonine Plague, literary sources must be examined to identify and subsequently model the potential disease responsible for this pandemic. The available literary evidence for the Antonine Plague consists of a description of the effects of the outbreak in ca. 189 CE recorded by the historian Dio (summarized, for example, by Duncan-Jones [20]). Literary evidence on the etiology of the Antonine Plague is based mainly on the writings of Galen (analyzed, for example, by Robert J. and M. L. Littmans [7], or, more recently, by Rebecca

Flemming [9]). The epidemiological characteristics of the modeled disease variants are based on recent analyses of the Antonine Plague [10, 21]. The demographic framework of the models is based on recent population estimates of the city of Rome [22]. Using the current understanding of the etiology of the Antonine Plague and the population estimates of the city of Rome in 165 CE as a starting point for building a series of dynamic ordinary differential equation models (ODE models henceforth) of disease spread, we were able to simulate the spread of the epidemic in Rome over the next 25 years and compare the output of the models with literary evidence of the impact of the Antonine Plague reported in historical sources. The aim of this comparison was to identify the most plausible causative agent of the Antonine Plague. On the basis of the description of symptoms in historical sources and a review of previous literature on the subject, we selected bubonic plague, bubonic plague with additional pneumonic transmission, smallpox, and measles as the most likely candidates.

The purpose of the study is to model the spread of several potential pathogens in the city of Rome during the Antonine Plague to identify the most plausible causative agent by comparing the simulated results with historical accounts and demographic data.

## Methods

### Model development

On the basis of the models developed by White and Mordechai [12], we adapted the following ODE models: (1) bubonic plague with the traditional rat, flea, and human dynamics (rat growth dynamics, innate resistance in rats, and variations with and without a compartment for exposed individuals in the incubation period, i.e., the Susceptible-Infected-Recovered and Susceptible-Exposed-Infected-Recovered variants [SIR and SEIR variants henceforth]); and (2) bubonic plague that can evolve into secondary pneumonic infection with further pneumonic transmission (only the SEIR variant). In addition to the models introduced by White and Mordechai, we developed two additional ODE models: (3) smallpox (SIR, SEIR) and (4) measles (SIR, SEIR). We compared time course results using expected parameter values (based on epidemiological literature) and compared the results from LHS sampling. We then compared the modeling results with literary evidence of the impact of the Antonine Plague on the city of Rome.

**Initial conditions and parameter estimates.**   Since there is no direct evidence concerning the demographic characteristics of the population of Rome in the 2nd century CE or the causative agent of the Antonine Plague, interpretations from demographic, archaeological, and epidemiological literature are crucial for the development of models that attempt to identify the most plausible cause of the Antonine Plague. The population of the city of Rome, traditionally estimated at about 1,000,000 individuals in 165 CE, has recently been re-estimated according to archaeological evidence of the habitable space of the city and the correlation between population density and urban space across history and cultures [22]. On the basis of the analysis by John W. Hanson and Scott G. Ortman, we adopted their revised population estimate of 923,406 individuals in the city of Rome in the mid-2nd century CE [22]. In addition, we set the birth rate and natural death rate to equal values and the average life expectancy to 25 years [23].

The models of plague variants were based on the literature on plague and adapted from those developed by White and Mordechai [12]. These models consisted of populations of rats, fleas, and humans, and the parameters of disease spread among them were derived from literature estimating the characteristics of documented plague outbreaks, disease models, and infection studies. All plague models except that of a pneumonic outbreak were adapted from White and Mordechai (the model of a purely pneumonic plague outbreak implies that rat and flea

population dynamics are not relevant to the progression of the epidemic; modeling a simultaneous pneumonic and bubonic outbreak was therefore considered more realistic). In addition, models were developed for the spread of smallpox and measles. The initial structure of the smallpox and measles models was derived from White and Mordechai's pneumonic plague model, but as with all plague models, certain demographic features (births and deaths from natural causes attributed to each compartment separately) and epidemiological mechanisms (the total population was calculated from the sum of all living compartments, not just their subset) were adapted. The purpose of these changes (calculating human births from the sum of all living compartments and calculating human deaths from natural causes for each compartment separately) was to ensure consistency among all disease models, as well as among specific parts of them (e.g., among equations describing rat population dynamics in the plague models). In addition to building the smallpox and measles models on the basis of the pneumonic plague models, the parameters for each disease were derived from epidemiological literature.

**Bubonic plague (humans, fleas, and rats).** The model of bubonic plague transmission, in which rodents and fleas play a crucial role, has been described by entomologists and in mathematical models [12, 24–26]. According to observations of plague outbreaks in India in the 20[th] century, it appears that rat fleas (*Xenopsylla cheopis*) may turn to humans as alternative hosts in situations of significant decline in black rat (*Rattus rattus*) populations [12]. For this reason, on the basis of previously developed models, we adapted three sets of equations describing the dynamics of plague progression through rat, flea, and human populations [12].

The dynamics of the disease spread in the rat population are given in the form of a Susceptible-Infected-Recovered (SIR) set of equations where the total number of rats ($N_r$) is given by:

$$N_r(t) = S_r(t) + I_r(t) + R_r(t)$$

The number of susceptible ($S_r$), infected ($I_r$), recovered ($R_r$), and dead ($D_r$) rats (affected by the number of free infectious fleas, F), when not taking into account the dynamics of their population growth, is given by:

$$\frac{dS_r}{dt} = -\beta_r S_r F(1 - e^{-\alpha N_r})/N_r$$

$$\frac{dI_r}{dt} = \beta_r S_r F(1 - e^{-\alpha N_r})/N_r - \gamma_r I_r$$

$$\frac{dR_r}{dt} = g_r \gamma_r I_r$$

$$\frac{dD_r}{dt} = (1 - g_r)\gamma_r I_r$$

The parameter $\beta_r$ is the transmission rate from rats to fleas, $\alpha$ is the flea searching efficiency, $\gamma_r^{-1}$ is the bubonic infectious period in rats, and $g_r$ is the probability of rats surviving bubonic plague infection [12]. Contact between rats and fleas is modeled as a "[r]andom search process of fleas within a limited area, which is modulated by the number of available rats ($N_r$) and the searching efficiency of fleas ($\alpha$)" (emphasis original [12]; see also [25–28]).

The movement of fleas according to the number of expected fleas per rat (*H*) and the number of free infectious fleas (*F*) in the environment that may encounter human hosts is described by the following equations (where the parameter $r_f$ is the birth rate of fleas, $K_f$ is flea

carrying capacity per rat, and $d_f^{-1}$ is the lifespan of fleas):

$$\frac{dH}{dt} = r_f H\left(1 - H/K_f\right)$$

$$\frac{dF}{dt} = (1 - g_r)\gamma_r I_r H - d_f F$$

The addition of population growth dynamics to the rat models was modified by the rat birth rate ($r_r$), capped by the carrying capacity of the environment ($K_r$) and limited by the death rate ($d_r$); there was also a probability of rats being born resistant to plague ($p_r$):

$$\frac{dS_r}{dt} = -r_r S_r(1 - N_r/K_r) + r_r R_r(1 - p_r) - d_r S_r - \beta_r S_r F(1 - e^{-\alpha N_r})/N_r$$

$$\frac{dI_r}{dt} = \beta_r S_r F(1 - e^{-\alpha N_r})/N_r - \gamma_r I_r$$

$$\frac{dR_r}{dt} = r_r R_r(p_r - N_r/K_r) + g_r\gamma_r I_r - d_r R_r$$

$$\frac{dD_r}{dt} = (1 - g_r)\gamma_r I_r$$

For the bubonic SIR model, the total number of humans ($N_h$) is given by:

$$N_h(t) = S_h(t) + I_h(t) + R_h(t)$$

The progression of bubonic plague through the compartments of humans who are susceptible ($S_h$), infected ($I_h$), recovered ($R_h$), dead due to disease ($D_h$), and dead due to age and other natural causes ($A_h$) is given by the following equations (where the parameter $\beta_b$ is the transmission rate of bubonic plague from rat fleas to humans, $b_h$ is the birth rate of humans, $d_h$ is the death rate of humans, $\gamma_b^{-1}$ is the duration of bubonic plague infectious period in humans, and $g_h$ is the probability of humans recovering from bubonic plague):

$$\frac{dS_h}{dt} = -\beta_b \frac{S_h}{N_h} F(e^{-\alpha N_r}) + b_h N_h - d_h S_h$$

$$\frac{dI_h}{dt} = \beta_b \frac{S_h}{N_h} F(e^{-\alpha N_r}) - \gamma_b I_h - d_h I_h$$

$$\frac{dR_h}{dt} = g_h\gamma_b I_h - d_h R_h$$

$$\frac{dD_h}{dt} = (1 - g_h)\gamma_b I_h$$

$$\frac{dA_h}{dt} = d_h N_h$$

The addition of the class of individuals exposed to bubonic plague but currently in the incubation period transforms the previous SIR model into a SEIR model (which is also reused from the models developed by White and Mordechai [12]). The total number of humans in the

bubonic plague SEIR model is described by the following equation:

$$N_h(t) = S_h(t) + E_h(t) + I_h(t) + R_h(t)$$

The number of humans who are susceptible ($S_h$), exposed ($E_h$), infected ($I_h$), recovered ($R_h$), dead due to disease ($D_h$), and dead due to age and other natural causes ($A_h$) in the bubonic SEIR model is given by the following equations (where the parameter $\sigma_b^{-1}$ is the duration of bubonic plague incubation period in humans and the remaining parameters are the same as for the bubonic plague SIR model in this section above):

$$\frac{dS_h}{dt} = -\beta_b \frac{S_h}{N_h} F(e^{-\alpha N_r}) + b_h N_h - d_h S_h$$

$$\frac{dE_h}{dt} = \beta_b \frac{S_h}{N_h} F(e^{-\alpha N_r}) - \sigma_b E_h - d_h E_h$$

$$\frac{dI_h}{dt} = \sigma_b E_h - \gamma_b I_h - d_h I_h$$

$$\frac{dR_h}{dt} = g_h \gamma_b I_h - d_h R_h$$

$$\frac{dD_h}{dt} = (1 - g_h)\gamma_b I_h$$

$$\frac{dA_h}{dt} = d_h N_h$$

The equations in the SEIR model account for different human subpopulations (susceptible, exposed, infected, recovered, dead from disease, and from natural causes). The transmission rate $\beta_b$ depends on the interaction between humans and the vector population, modeled by $\beta_b S_h F(e^{-\alpha N_r})/N_h$, which describes how rat fleas transmit plague to susceptible humans. Positive effects include human births ($b_h N_h$) and recovery from infection ($g_h \gamma_b I_h$), while negative effects include infection and disease progression ($\sigma_b E_h$), death due to infection ($(1-g_h)\gamma_b I_h$), and natural death ($d_h$ in all subpopulations). The subpopulation of exposed individuals ($E_h$) slows down the progression of plague compared to the simpler SIR model.

**Bubonic/pneumonic transmission.** Since a plausible scenario for the spread of disease in the city of Rome during the Antonine Plague is the parallel spread of a bubonic form of the plague along with its pneumonic variant, in which the disease in an individual progresses from a bubonic variant to a secondary respiratory infection, the combined model developed by White and Mordechai was reused [12, 29, 30]. For this scenario, only the SEIR model was developed; the equations defining the dynamics in the rat and flea populations remain the same, and the total number of humans is described as follows:

$$N_h(t) = S_h(t) + E_b(t) + E_p(t) + I_b(t) + I_p(t) + R_h(t)$$

As noted in the equation above for the total human population, individuals can be exposed to a different plague variant (bubonic or pneumonic, becoming $E_b$ or $E_p$). They may, in turn, either recover ($R_h$) or become infectious and start spreading the respective variant they contracted (either the bubonic or the pneumonic plague variant, becoming $I_b$ or $I_p$). Following the SIR/SEIR models of bubonic plague transmission described above, the parameter $\beta_b$ is the

transmission rate of bubonic plague from rat fleas to humans, $\beta_p$ is the pneumonic plague transmission rate in humans, $b_h$ is the birth rate of humans, $d_h$ is the death rate of humans, $\sigma_b^{-1}$ is the duration of bubonic plague incubation period in humans, $\gamma_b^{-1}$ is the duration of bubonic plague infectious period in humans, $\gamma_p^{-1}$ is the duration of pneumonic plague infection period in humans, $p$ is the probability of human bubonic plague developing into secondary pneumonic plague, and $g_h$ is the probability of humans recovering from bubonic plague:

$$\frac{dS_h}{dt} = -\beta_b \frac{S_h}{N_h} F(e^{-\alpha N_r}) - \beta_p S_h \frac{I_h}{N_h} + b_h N_h - d_h S_h$$

$$\frac{dE_b}{dt} = \beta_b \frac{S_h}{N_h} F(e^{-\alpha N_r}) - \sigma_b E_b - d_h E_b$$

$$\frac{dE_p}{dt} = \beta_b S_h \frac{I_p}{N_h} - \sigma_p E_p - d_h E_p$$

$$\frac{dI_b}{dt} = \sigma_b E_b - \gamma_b I_b - d_h I_b$$

$$\frac{dI_p}{dt} = \sigma_p E_p + p\gamma_b I_b - \gamma_p I_p - d_h I_p$$

$$\frac{dR_h}{dt} = g_h \gamma_b I_b - d_h R_h$$

$$\frac{dD_h}{dt} = (1 - p - g_h)\gamma_b I_b + \gamma_p I_p$$

$$\frac{dA_h}{dt} = d_h N_h$$

This model describes the concurrent spread of bubonic and pneumonic plague. Humans can either contract the bubonic variant via fleas or directly acquire the pneumonic variant from infected humans. The parameters $\beta_b$ and $\beta_p$ account for these processes, where $\beta_b S_h F(e^{-\alpha N_r})/N_h$ models flea-borne transmission, and $\beta_p S_h I_h/N_h$ represents human-to-human transmission of the pneumonic variant. Positive effects include recovery ($g_h \gamma_b I_b$) and births ($b_h N_h$), while negative effects include infection and disease progression ($\sigma_b E_b$, $\sigma_p E_p$), death due to infection (($1-p-g_h)\gamma_b I_b + \gamma_p I_p$), and natural death ($d_h$ in all subpopulations). The interaction between the variants is captured by the term $p\gamma_b I_b$, representing the progression from bubonic to pneumonic infection.

**Smallpox (humans only).** Since other diseases besides bubonic plague and its pneumonic variant are considered as possible causes of the Antonine Plague, smallpox and measles were developed in addition to the plague models. The framework for their development was based on the pneumonic plague SIR and SEIR models developed by White and Mordechai [12]. The total number of humans ($N_h$) is as follows:

$$N_h(t) = S_h(t) + I_h(t) + R_h(t)$$

Furthermore, the equations describing the movement among the susceptible ($S_h$), infected ($I_h$), removed ($R_h$), dead due to disease ($D_h$), and dead due to age and other natural causes ($A_h$)

compartments are as follows (where the parameter $\beta_s$ is the smallpox transmission rate in humans, $b_h$ is the birth rate of humans, $d_h$ is the death rate of humans, $\gamma_s^{-1}$ is the duration of smallpox infection period in humans, and $g_s$ is the probability of humans recovering from smallpox):

$$\frac{dS_h}{dt} = b_h N_h - \beta_s S_h \frac{I_h}{N_h} - d_h S_h$$

$$\frac{dI_h}{dt} = \beta_s S_h \frac{I_h}{N_h} - \gamma_s I_h - d_h I_h$$

$$\frac{dR_h}{dt} = g_s \gamma_s I_h - d_h R_h$$

$$\frac{dD_h}{dt} = (1 - g_s) \gamma_s I_h$$

$$\frac{dA_h}{dt} = d_h N_h$$

The SIR model describes the spread of smallpox among humans, where the total population consists of susceptible ($S_h$), infected ($I_h$), and recovered ($R_h$) individuals. The transmission rate of smallpox is governed by $\beta_s$, which reflects the rate at which susceptible individuals become infected trough contact with the infected population. The positive effects to the population growth consist of birth rate ($b_h N_h$), while negative effects include transmission of the disease and death from natural causes ($d_h$). Infected individuals either recover ($g_s \gamma_s I_h$) or die from the disease (($1-g_h$)$\gamma_b I_h$), and natural deaths affect all compartments. This model captures the direct transitions between susceptible, infected, and recovered humans, simplifying smallpox dynamics by excluding an incubation period.

Extending this to a SEIR framework by adding in an incubation period ($\sigma_s^{-1}$), the total number of humans becomes:

$$N_h(t) = S_h(t) + E_h(t) + I_h(t) + R_h(t)$$

In turn, the equations describing the dynamics of humans moving among the susceptible ($S_h$), exposed ($E_h$), infected ($I_h$), removed ($R_h$), dead due to disease ($D_h$), and dead due to age and other natural causes ($A_h$) compartments are as follows:

$$\frac{dS_h}{dt} = b_h N_h - \beta_s S_h \frac{I_h}{N_h} - d_h S_h$$

$$\frac{dE_h}{dt} = \beta_s S_h \frac{I_h}{N_h} - \sigma_s E_h - d_h E_h$$

$$\frac{dI_h}{dt} = \sigma_s E_h - \gamma_s I_h - d_h I_h$$

$$\frac{dR_h}{dt} = g_s \gamma_s I_h - d_h R_h$$

$$\frac{dD_h}{dt} = (1 - g_s)\gamma_s I_h$$

$$\frac{dA_h}{dt} = d_h N_h$$

In the SEIR model, the smallpox dynamics are expanded to include an incubation period ($E_h$) before individuals become infectious. Susceptible individuals ($S_h$) can be exposed to smallpox through contact with the infectious population, with transmission governed by $\beta_s$. After an incubation period ($\sigma_s^{-1}$), exposed individuals ($E_h$) transition into the infected compartment ($I_h$). As in the SIR model, infected individuals can either recover ($g_s\gamma_s I_h$) or die from the disase ($(1-g_s)\gamma_s I_h$), and death from natral causes affects all compartments. This model adds realism by accounting for the delay between exposure and infectiousness, reflecting the incubation period of smallpox.

**Measles (humans only).**   Since the mode of transmission of measles, another candidate for the probable cause of the Antonine Plague, is similar to that of smallpox, the equations are virtually identical, differing only in the parameters that define the model's behavior. The total number of humans ($N_h$) for a simple SIR model is given by the following equation:

$$N_h(t) = S_h(t) + I_h(t) + R_h(t)$$

The equations describing the dynamics of humans moving among the susceptible ($S_h$), infected ($I_h$), removed ($R_h$), dead due to disease ($D_h$), and dead due to age and other natural causes ($A_h$) compartments are as follows (where the parameter $\beta_m$ is the measles transmission rate in humans, $b_h$ is the birth rate of humans, $d_h$ is the death rate of humans, $\gamma_m^{-1}$ is the duration of measles infection period in humans, and $g_m$ is the probability of humans recovering from measles):

$$\frac{dS_h}{dt} = b_h N_h - \beta_m S_h \frac{I_h}{N_h} - d_h S_h$$

$$\frac{dI_h}{dt} = \beta_m S_h \frac{I_h}{N_h} - \gamma_m I_h - d_h I_h$$

$$\frac{dR_h}{dt} = g_m \gamma_m I_h - d_h R_h$$

$$\frac{dD_h}{dt} = (1 - g_m)\gamma_m I_h$$

$$\frac{dA_h}{dt} = d_h N_h$$

The equations describing the total human population in the SEIR measles model and the dynamics of humans moving among the susceptible ($S_h$), exposed ($E_h$), infected ($I_h$), removed ($R_h$) dead due to disease ($D_h$), and dead due to age and other natural causes ($A_h$) compartments are as follows (where $\sigma_m^{-1}$ is the duration of measles incubation period in humans and the remaining parameters are the same as for the smallpox SIR model in this section above):

$$N_h(t) = S_h(t) + E_h(t) + I_h(t) + R_h(t)$$

$$\frac{dS_h}{dt} = b_h N_h - \beta_m S_h \frac{I_h}{N_h} - d_h S_h$$

$$\frac{dE_h}{dt} = \beta_m S_h \frac{I_h}{N_h} - \sigma_m E_h - d_h E_h$$

$$\frac{dI_h}{dt} = \sigma_m E_h - \gamma_m I_h - d_h I_h$$

$$\frac{dR_h}{dt} = g_m \gamma_m I_h - d_h R_h$$

$$\frac{dD_h}{dt} = (1 - g_m) \gamma_m I_h$$

$$\frac{dA_h}{dt} = d_h N_h$$

Both measles SIR and SEIR models are like the smallpox models based on pneumonic plague SIR and SEIR models developed by White and Mordechai [12], therefore the general effects of variables affecting population growth and the spread of disease are for each of these diseases the same.

**Baseline mortality (humans only, no specific disease).** To account for the increase in mortality, we created a very simple model of the Roman population. This model does not take into account any specific disease, and its input parameters are only birth rate ($b_h$) and death rate ($d_h$), in addition to an initial size of the population of the city of Rome ($S_h$). The output of the model is the total number of dead individuals due to age and other natural causes ($A_h$):

$$\frac{dS_h}{dt} = b_h S_h - d_h S_h$$

$$\frac{dA_h}{dt} = d_h S_h$$

This simple model describes the population dynamics by focusing on a single compartment of susceptible individuals ($S_h$). The model assumes that the population changes only due to births ($b_h$) and natural deaths ($d_h$). By tracking the cumulative number of deaths in the $A_h$ compartment, we can estimate the total number of deaths over time. This allows for comparison between the population's baseline mortality (in the absence of disease) and the effects of disease-related mortality, as modeled in the other scenarios. Table 1 contains an overview of parameters used to develop all of the models in this study.

## Time course results and sensitivity analysis

Adapting the framework of White and Mordechai [12], we solved each model numerically using the *ode* function of the *deSolve* R package [57]. The plague models were solved with several variations of the ratio of rats to humans (1:2, 2:1, 1:1) to see whether this initial condition influences the models' behavior at the end of the modeled period. Using Latin Hypercube Sampling with Partial Rank Correlation Coefficients (LHS-PRCC), we performed a sensitivity analysis to determine the importance of each parameter on the modeled behavior [12, 58]. The

**Table 1. Expected parameter values derived from literature (plague-related parameters adapted from White and Mordechai).**

| Parameter | Description | Expected value [Range: min, max] | Distribution | Reference(s) |
|---|---|---|---|---|
| $b_h$ | Human birth rate | $1/(25 \cdot 365)$ $[1/(30 \cdot 365), 1/(20 \cdot 365)]$ days$^{-1}$ | Uniform | [12, 23, 25] |
| $d_h$ | Human death rate due to age and other natural causes | $1/(25 \cdot 365)$ $[1/(30 \cdot 365), 1/(20 \cdot 365)]$ days$^{-1}$ | Uniform | [12, 23, 25] |
| $\beta_p$ | Pneumonic plague transmission rate in humans | 0.08 [0.01, 1] days$^{-1}$ | Uniform | [12, 27, 31–33] |
| $\sigma_p^{-1}$ | Duration of pneumonic plague incubation period in humans | 4.3 [2.5, 6.1] days | Normal | [12, 31, 34] |
| $\gamma_p^{-1}$ | Duration of pneumonic plague infection period in humans | 2.5 [1.3, 3.7] days | Normal | [12, 31] |
| $\beta_r$ | Transmission rate of plague from fleas to rats | 1.248 [0, 3.67] fleas$^{-1}$ days$^{-1}$ | Triangle | [12, 27, 35] |
| $\alpha$ | Flea searching efficiency | $3/S_r$ (t = 0) [0.39 < $\alpha K_r$ < 20] rats$^{-1}$ | Uniform | Varied across the full range of $S_r$, $K_r$, and $\alpha$ values in the plague models and through sensitivity analysis to examine their combined impact on the model's behavior [12, 25, 36] |
| $r_r$ | Rat birth rate | 0.014 [0.011, 0.016] days$^{-1}$ | Uniform | [12, 25, 26] |
| $K_r$ | Carrying capacity of rats | $N_r$ (t = 0) [0.5$N_r$ (t = 0), 1.5$N_r$ (t = 0)] rats | Uniform | Varied across the full range of $N_r$ (t = 0) and $K_r$ values through sensitivity analysis to examine their combined impact on the model's behavior [12] |
| $d_r$ | Rat death rate due to age and other natural causes | 0.2/365 [0.1/365, 0.3/365] days$^{-1}$ | Uniform | [12, 25, 26] |
| $p_r$ | Probability of rats inheriting resistance to plague | 0.65 [0.4, 0.9] | Uniform | [12, 37–39] |
| $\gamma_r^{-1}$ | Duration of bubonic plague infectious period in rats | 5.15 [4.71, 5.59] days | Normal | [12, 38] |
| $g_r$ | Probability of rats recovering from bubonic plague | 0.06 [0.0, 0.37] | Triangle | [12, 38] |
| $r_f$ | Flea birth rate | 0.0084 [0.0084, 0.055] days$^{-1}$ | Uniform | [12, 25–27] |
| $K_f$ | Flea carrying capacity per rat | 6 [3.29, 11.17] fleas | Normal | [12, 25–27] |
| $d_f^{-1}$ | Flea lifespan | 5 [1, 11.66] days | Triangle | [12, 27, 40] |
| $\beta_b$ | Transmission rate of bubonic plague from rat fleas to humans | 0.19 [0.01, 1] days$^{-1}$ | Uniform | [12, 27] |
| $\sigma_b^{-1}$ | Duration of bubonic plague incubation period in humans | 4 [2, 6] days | Triangle | [12, 29] |
| $\gamma_b^{-1}$ | Duration of bubonic plague infectious period in humans | 10 [3, 10] days | Triangle | [12, 27, 29, 30] |
| $g_h$ | Probability of humans recovering from bubonic plague | 0.34 [0.3, 0.4] | Triangle | [12, 27, 41] |
| $p$ | Probability of human bubonic plague developing into secondary pneumonic plague | 0.1 [0, 0.15] | Triangle | [12, 29, 30, 34, 42–44] |
| $\beta_s$ | Smallpox transmission rate in humans | 0.584 [0.584, 0.6] days$^{-1}$ | Uniform | [45, 46] |
| $\sigma_s^{-1}$ | Duration of smallpox incubation period in humans | 12 [7, 17] days | Normal | [47, 48] |
| $\gamma_s^{-1}$ | Duration of smallpox infectious period in humans | 9.5 [8, 11] days | Normal | [47] |
| $g_s$ | Probability of humans recovering from smallpox | 0.1 [0.05, 0.7] | Triangle | [47] |

*(Continued)*

**Table 1.** (Continued)

| Parameter | Description | Expected value [Range: min, max] | Distribution | Reference(s) |
|---|---|---|---|---|
| $\beta_m$ | Measles transmission rate in humans | 1.175 [0.888, 1.333] days$^{-1}$ | Uniform | Assumed from $R_0 = \beta/\gamma$ [49, 50]; since $R_{m0}$ = [12, 18] [50] and $\gamma_m$ = 13.5 [51], $\beta_m$ = [0.888, 1.333] days$^{-1}$; 400 year$^{-1}$ [52]; 1.175 days$^{-1}$ [53] |
| $\sigma_m^{-1}$ | Duration of measles incubation period in humans | 10 [9, 11] days | Normal | [54] |
| $\gamma_m^{-1}$ | Duration of measles infectious period in humans | 13.5 [13, 14] days | Normal | 13 days [51]; 2 weeks [52] |
| $g_m$ | Probability of humans recovering from measles | 0.7 [0.66, 0.97] | Triangle | Case fatality rate (CFR) when unvaccinated is between 1–3% and 10–30% [55]; CFR is 3–34% in developing countries [56] |

complexity of our models ranges from 5 parameters in the simplest respiratory SIR ones (smallpox and measles) to 17 parameters in the bubonic plague models, which take into account the carrying capacity of the environment to limit the maximum number of rats. Consistent with the White and Mordechai paper [12], we used the *lhs* package in R (4.1.2) to create an LHS framework of 100 subdivisions per parameter [58, 59].

We performed PRCC analysis on the modeled total mortality and outbreak duration. For the purpose of this paper and in accordance with White and Mordechai [12], we considered as outbreak days only those on which the mortality rate due to disease exceeded 100 deaths per day (this number is not arbitrary but reflects the expected baseline mortality due to natural, non-pandemic causes in the city of Rome, based on the behavior of our models when no disease was introduced). We used Bonferroni corrected p-value to compute confidence intervals and the *pcc* function of the *sensitivity* package to compute the PRCC values with 500 bootstrap replicates [12, 60]. Our code is deposited on Zenodo: https://zenodo.org/doi/10.5281/zenodo.13857741.

## Model evaluation: Evidence from primary and secondary sources

Scholarly interpretations of the impact of the Antonine Plague vary widely, mainly because the primary sources do not provide much quantitative information about its characteristics. Galen vaguely reports large-scale mortality in the army due to disease in Aquileia during the winter (possibly in 168/169) but does not elaborate beyond saying that most individuals died [20, 61]. Also, Galen´s information that he lost almost all the slaves in his household provides very little insight into the severity of the Antonine Plague [9].

However, the more quantitative estimate reported by Cassius Dio provides insight into how many people might have been dying around the year 189 CE, possibly in the last years of the pandemic (Dio Cassius 73:14.3–4, [62]):

"[M]oreover, a pestilence occurred, the greatest of any of which I have knowledge; for two thousand persons often died in Rome in a single day."

Despite its brevity, this mortality estimate, usually interpreted as referring to the Antonine Plague [7], provides an invaluable insight into the impact of the disease on a single location at the time of Commodus' reign. If we suppose that these mortality figures refer to a late stage of the Antonine Plague (and assume that they are not simply a rhetorical exaggeration typical of disease accounts in antiquity), it should be reasonable to expect a similar mortality pattern to emerge in appropriately modeled scenarios of disease spread in Rome.

In addition to explicit observations in primary sources, interpretations of writings and archaeological material show attempts to provide more specific numbers regarding the overall impact of the Antonine Plague. According to Otto Seeck, over half of the Empire's population

died during this pandemic [63]. In contrast, J. F. Gilliam concluded that only 1–2% beyond ordinary mortality (or 500,00 to 1,000,000 individuals) died due to this plague [5]. The Littmans concluded that the average death rate in the Empire was 7–10% and perhaps 13–15% in cities and the army [7]. According to D. W. Rathbone's comparison with the 14th-century Black Death in Europe, the Antonine Plague may have caused a 20–30% decline in the population of Egypt [64]. According to Walter Scheidel, the overall mortality was in the order of 25% [3]. William V. Harris argued that the mortality in the first wave of the pandemic was 16–22% and during the second wave 12–16%, amounting to over 16–20 million deaths from disease [65]. Harris's estimates were largely based on the work of Y. Zelener, who developed a mathematical model of the spread of smallpox and argued that, depending on estimates of the Empire's population and its post-pandemic fertility response, the mortality would have been around 22–24%, perhaps exceeding 25% in the case of a low fertility response [10]. One of the most recent estimates is presented by Kyle Harper, who argued that the worst affected areas experienced an increase in mortality of around 20%, while the overall effect would have been an increase in mortality of between 8 and 10% in the entire Roman Empire [21, 66].

## Results

### Model outcomes with expected parameter values

The results of the plague models range from ca. 581,000–589,000 mortalities (63–64% of the city's population in 165 CE) for the simplest bubonic ones to ca. 582,000 for the bubonic-pneumonic one. Similarly to the results of White and Mordechai [12], the variants of the bubonic plague model that took into account rat population dynamics showed a higher mortality rate, ca. 763,000–764,000 deaths due to disease (82–83% of the population of Rome). The smallpox models resulted in even higher mortality of ca. 881,000 deaths due to disease (95% of the population). On the other hand, the measles models indicated fewer fatalities, ca. 433,000–434,000 deaths (46–47% of Rome's population). Apart from differences in the number of mortalities due to disease, all of the models followed a similar pattern, with one large initial wave of disease spread and subsequent deaths, which was followed by one or two much smaller waves later on in the pandemic; these smaller waves following the initial one appear insignificant in comparison to the impact indicated by the first wave.

A mortality of over 100 deaths per day was reached between the 14th and 16th days of the outbreak in the plague models (lasting from 77 to 218 days), between the 16th and 62nd days in the smallpox models (lasting from 204 to 251 days), and between the 9th and 36th days in the measles models (lasting from 184 to 209 days). The possibly more noticeable death rate of 250 lives a day was reached in the plague models between the 15th and 17th days (lasting from 67 to 72 days), between the 17th and 69th days in the smallpox models (lasting from 111 to 144 days), and between the 9th and 40th days in the measles models (lasting from 86 to 88 days). The significant mortality we were interested in, at least 2,000 deaths per day, was reached between the 18th and 28th days of the outbreak in the plague models (lasting from 43 to 46 days), between the 22nd and 86th days in the smallpox models (lasting from 44 and 76 days), and between the 11th and 49th days in the measles models (lasting from 34 to 47 days). To see what the overall increase in mortality might have been, we also modeled a scenario without any specific disease, which resulted in a total of 960,696 dead individuals by the end of the modeled period. In turn, we compared the sum of individuals dead due to disease and dead due to age and other natural causes with the baseline value of 960,696 deaths to see how the modeled scenarios correspond with interpretations of the impact of the Antonine Plague in secondary literature.

Table 2 summarizes the results (detectable outbreak duration, maximal daily mortality rate, and total mortality due to disease) of a simulation starting a) with the same number of rats and

**Table 2. Summary of model output for each model (with rat to human ratio of 1:1 for the plague models).**

| Model | Detectable outbreak duration (deaths/day) | | | Maximal daily mortality rate due to disease (deaths/day) | Total mortality due to disease (humans) | Total mortality due to age and other natural causes (humans) | Total mortality increase (compared to the modeled baseline of 960696 deaths; percent) |
|---|---|---|---|---|---|---|---|
| | > 100 | > 250 | > 2000 | | | | |
| Bubonic SIR | 77 | 67 | 43 | 28011 | 589158 | 349926 | -2.24 |
| Bubonic SEIR | 80 | 70 | 46 | 23810 | 580881 | 358769 | -2.19 |
| Bubonic SIR (Rat dynamics) | 130 | 68 | 43 | 27946 | 763554 | 241917 | 4.66 |
| Bubonic SEIR (Rat dynamics) | 218 | 72 | 46 | 23759 | 763411 | 242563 | 4.71 |
| Bubonic-pneumonic SEIR | 81 | 71 | 46 | 23592 | 582222 | 357405 | -2.19 |
| Smallpox SIR | 204 | 111 | 44 | 48451 | 880359 | 78622 | -0.17 |
| Smallpox SEIR | 251 | 144 | 76 | 20103 | 880971 | 87800 | 0.84 |
| Measles SIR | 184 | 80 | 34 | 15614 | 433754 | 594499 | 7.02 |
| Measles SEIR | 209 | 88 | 47 | 7822 | 432658 | 597256 | 7.20 |

humans for the plague models (with one of the rats being infected), i.e., $S_h(t = 0) = 923,406$, $S_r(t = 0) = 923,405$, and $I_r(t = 0) = 1$ (Fig 1); and b) no rats and 1 infected human for the smallpox and measles models, i.e., $S_h(t = 0) = 923,406$ and $I_h(t = 0) = 1$.

When we changed the ratio of rats to humans, we observed similar changes as described by White and Mordechai [12]–in our case, more pronounced due to a larger initial susceptible population of both rats and humans.

With a 1:2 ratio of rats to humans, the results of the plague models range from ca. 461,000–475,000 mortalities (ca. 50% of the population) for the simplest bubonic ones to ca. 467,000 for the bubonic-pneumonic one (Table 3 and S1 Fig). The models that took into account rat population dynamics indicated ca. 761,000 deaths (82% of the population). In this scenario, the duration of the simpler bubonic and bubonic-pneumonic plague models ranged from 78 to 82 days with over 100 deaths per day, 68 to 71 days with over 250 deaths per day, and 43 to 47 days with over 2,000 deaths each day. The durations of the models with rat population dynamics were much longer– 589–626 days with over 100 daily deaths, 280–312 days with over 250 daily deaths, and 44–46 days with over 2,000 daily deaths. In all models in the 1:2 scenario, the highest death count in a day was in the range of ca. 17,200–19,700.

With a 2:1 ratio of rats to humans, the results of the plague models range from ca. 609,000 deaths (ca. 66% of the population) in the bubonic and bubonic-pneumonic models to ca. 764,000 deaths due to disease (ca. 82% of the population) in the plague models with rat population dynamics (Table 4 and S2 Fig). With such a rat-to-human population ratio, the duration of all plague models was nearly the same, ranging from 74 to 77 days with over 100 mortalities per day, 63 to 67 days with over 250 mortalities each day, and 40 to 44 days with over 2,000 mortalities each day. The maximal daily deaths were in the range of ca. 27,900–34,700.

## Sensitivity analysis: Model outcome variability and parameter influence

In the basic bubonic plague SIR and SEIR models without rat population dynamics, flea searching efficiency ($\alpha$), flea death rate ($d_f$), and rat recovery probability ($g_r$) were negatively correlated with outbreak size (number of deaths) (S4 Fig, panels A & C) and outbreak duration (S4 Fig, panel B & D). In the bubonic plague SIR and SEIR models with rat population dynamics, flea searching efficiency ($\alpha$) was negatively correlated with outbreak size, and the

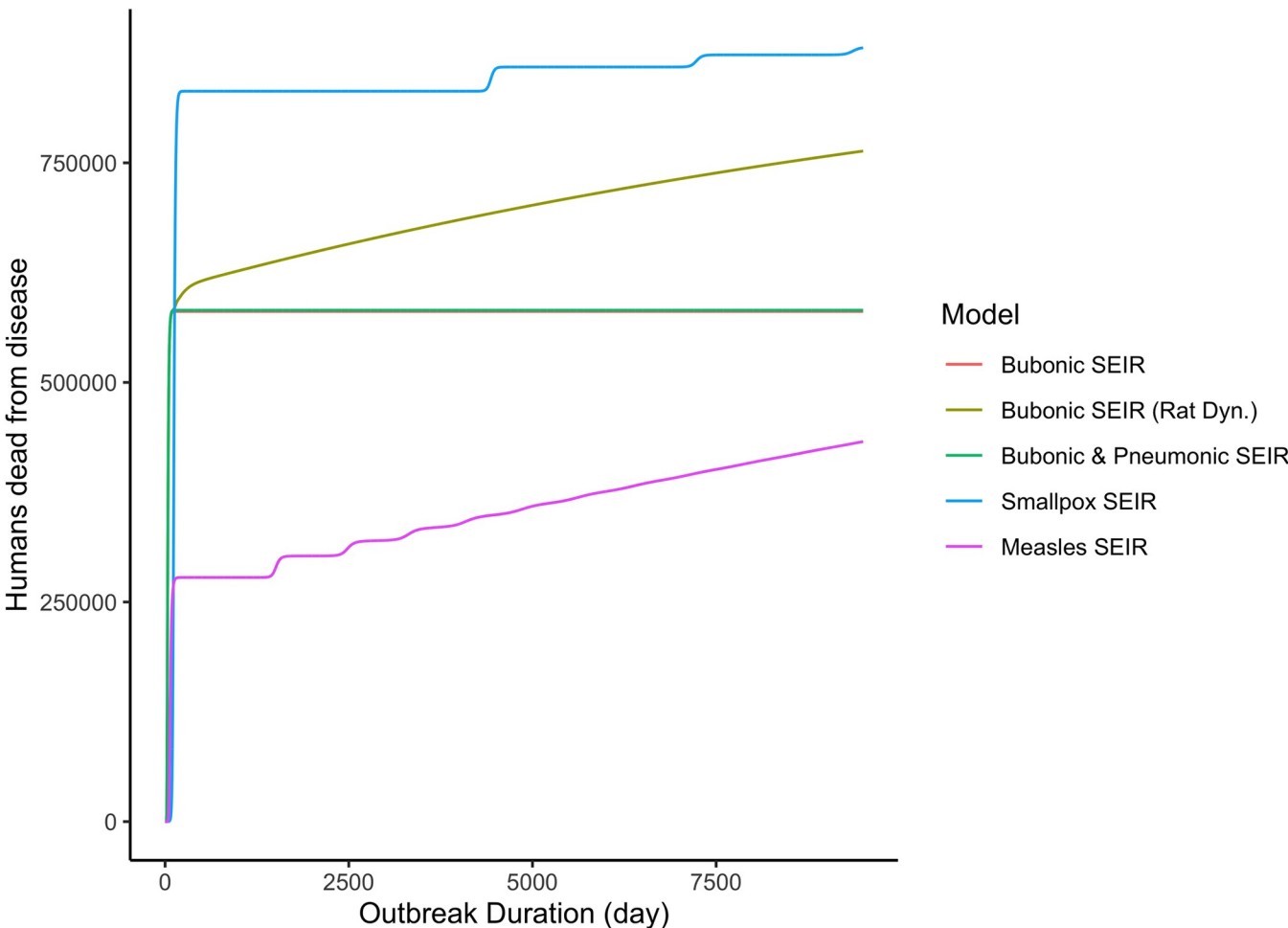

**Fig 1. Time course results of different transmission mode models.** In the plague models with a rat to human ratio of 1:1).

transmission rate from fleas to humans ($\beta_b$) was moderately positively correlated with outbreak size (S5 Fig, panels A & C). In the bubonic-pneumonic plague SEIR model, the duration of the pneumonic infection period in humans ($\gamma_p^{-1}$) was negatively correlated with outbreak size, the duration of the pneumonic incubation period in humans ($\sigma_p^{-1}$) was positively

**Table 3. Summary of model output for the plague models with rat to human ratio of 1:2.**

| Model | Detectable outbreak duration (deaths/day) | | | Maximal daily mortality rate due to disease (deaths/ day) | Total mortality due to disease (humans) | Total mortality due to age and other natural causes (humans) | Total mortality increase (compared to the modeled baseline of 960696 deaths; percent) |
|---|---|---|---|---|---|---|---|
| | > 100 | > 250 | > 2000 | | | | |
| Bubonic SIR | 78 | 68 | 43 | 19740 | 474822 | 468542 | -1.80 |
| Bubonic SEIR | 81 | 71 | 46 | 17301 | 461560 | 482489 | -1.73 |
| Bubonic SIR (Rat dynamics) | 589 | 280 | 44 | 19662 | 761388 | 250196 | 5.29 |
| Bubonic SEIR (Rat dynamics) | 626 | 312 | 46 | 17235 | 761163 | 251387 | 5.39 |
| Bubonic-pneumonic SEIR | 82 | 71 | 47 | 17217 | 466546 | 477348 | -1.74 |

**Table 4. Summary of model output for the plague models with rat to human ratio of 2:1.**

| Model | Detectable outbreak duration (deaths/day) | | | Maximal daily mortality rate due to disease (deaths/day) | Total mortality due to age and other natural causes (humans) | Total mortality due to age and other natural causes (humans) | Total mortality increase (compared to the modeled baseline of 960696 deaths; percent) |
|---|---|---|---|---|---|---|---|
| | > 100 | > 250 | > 2000 | | | | |
| Bubonic SIR | 74 | 63 | 40 | 34697 | 609326 | 328879 | -2.34 |
| Bubonic SEIR | 77 | 67 | 44 | 28342 | 608675 | 329832 | -2.30 |
| Bubonic SIR (Rat dynamics) | 74 | 64 | 40 | 34659 | 764272 | 240192 | 4.55 |
| Bubonic SEIR (Rat dynamics) | 78 | 67 | 44 | 28315 | 764160 | 240630 | 4.58 |
| Bubonic-pneumonic SEIR | 77 | 67 | 43 | 27908 | 608937 | 329586 | -2.30 |

correlated with outbreak size (S6 Fig, panel A), flea searching efficiency ($\alpha$) was moderately positively correlated with outbreak duration, and the transmission rate of bubonic plague from fleas to humans ($\beta_b$) was negatively correlated with outbreak duration (S6 Fig, panel B). The sensitivity analysis of plague models was consistent with the results of White and Mordechai [12].

In the smallpox SIR and SEIR models, the birth rate ($b_h$) was positively correlated with outbreak size, the natural death rate ($d_h$) was moderately negatively correlated with outbreak size (the correlation was stronger in the SIR model), and the probability of recovering from smallpox ($g_s$) was strongly negatively correlated with outbreak size (S7 Fig, panels A & C). Furthermore, the birth rate ($b_h$) and the probability of recovering from smallpox ($g_s$) were positively correlated with outbreak duration (S7 Fig, panels B & D). In addition, the duration of the smallpox incubation period ($\sigma_s^{-1}$) was moderately negatively correlated with outbreak duration in the SEIR model (S7 Fig, panel D).

In the measles SIR and SEIR models, the birth rate ($b_h$) was strongly positively correlated with outbreak size, and the natural death rate ($d_h$) and the probability of recovering from measles ($g_m$) were strongly negatively correlated with outbreak size (S8 Fig, panels A & C). Moreover, the transmission rate of measles ($\beta_m$) and the probability of recovering from measles ($g_m$) were strongly negatively correlated with outbreak duration (S8 Fig, panels B & D).

The sensitivity analysis was consistent across both non-uniformly and uniformly distributed ranges of parameter values and is described in detail in the appendices (S1 and S2 Appendices).

## Discussion

Current debate about the cause(s) of the Antonine Plague has not yielded much insight into the nature of the pathogen(s) concerned but has suggested several candidates based on our knowledge of pathogens encountered in modern history. Despite the lack of more precise data on the cause of this pandemic, scholars have come up with different interpretations of the magnitude of its impact. We adapted and developed several models of disease spread to see if more light could be shed on the progression of the Antonine Plague in the city of Rome.

### Bubonic plague

Bubonic plague models resulted in 581,000–764,000 deaths due to disease when the rat count was equal to the number of humans. In the scenario with a 1:2 ratio of rats to humans, the mortalities due to disease varied between 461,000–761,000, and in the scenario with a 2:1 ratio

of rats to humans, 609,000–764,000 died due to plague. The extent of such dying is massive (ca. 50–80% of the initial population of the city), but none of these models provided any indication of multiple significant waves. In particular, almost no mortality due to disease was indicated by the end of the modeled period.

Despite the fact that the overall count of deaths may seem incredibly large, when natural deaths, i.e., deaths not caused by the bubonic plague, are taken into account, it turns out that the total number of deaths was actually about 2% lower in the models that didn't calculate rat population dynamics, and about 5% higher in the models that did so (compared to the expected demographic development of Rome calculated by our baseline model). What seems clear even from our models, however, is that the number of rats plays a dominant role in the transmission of plague, driving the progression of the outbreak [25].

## Combined bubonic-pneumonic plague

Bubonic-pneumonic plague models resulted in 582,000 deaths due to disease in the scenario with the same number of rats and humans and ranged from about 115,000 fewer deaths in the scenario with a 1:2 ratio of rats to humans to about 27,000 more deaths in the scenario with a 2:1 ratio of rats to humans. This result is only slightly higher than the outcome of the bubonic SEIR model without rat population dynamics (50–60% of Rome's initial population), but although it appears to be almost as massive as the outcomes of the other plague models, it indicates an overall decrease in mortality of about 2% (when deaths from natural, non-plague causes are also considered). Because of the similarity of the progression of the combined bubonic-pneumonic plague models to that of the bubonic plague models, it is not possible to conclude which of these transmission routes is more plausible as the cause of the Antonine Plague.

## Smallpox

Smallpox models showed about 881,000 deaths due to disease by the end of the modeled period (95% of the initial population), which is the highest death toll from all the present models. Unlike the plague models, the smallpox ones depend only on humans for transmission, smallpox thus burning out rather quickly in one large wave (which was, however, comparable to that emerging in bubonic plague models with rat population dynamics). Taking into account deaths due to non-smallpox causes, the resulting mortality rate was almost 0.2% lower in the SIR model and nearly 1% higher in the SEIR one when compared to the modeled baseline mortality. Sensitivity analysis showed that the birth rate ($b_h$) was positively correlated with the outbreak size and duration of the outbreak, and that the death rate ($d_h$) was negatively correlated with the size of the outbreak, underscoring the main driver of the disease, i.e., humans (in contrast to the leading role of rats in the plague models). The overall decrease in mortality by the end of the modeled period might be caused by the large number of deaths at the beginning of the outbreak, after which the disease burned out of susceptible hosts and led to much slower demographic development afterward (resulting in a much lower count of naturally-caused deaths during the whole period).

## Measles

The measles models showed a mortality of about 434,000 deaths due to the disease (ca. 47% of the initial population), which is the lowest death count indicated by any of the models in this article. Like smallpox, measles requires only humans for transmission, so the dynamics of disease progression were much less complex than in the plague models. Even though the number of deaths due to this disease was much lower than the numbers due to smallpox or plague, the

overall impact on the city of Rome was much more significant: due to the lower impact on the population of Rome, measles was likely able to leave a larger pool of humans capable of reproducing and, in turn, lead to more individuals susceptible to infection in the long run. As a result, when combined with naturally occurring deaths, the impact of measles resulted in approximately 7% more deaths than if no pathogen was introduced into the city of Rome.

## Limitations

The models presented in this article use parameters recorded or estimated in modern history. However, it is essential to emphasize that the specific disease that caused the Antonine Plague may have been an ancestor of today's known pathogens and, therefore, may have behaved differently. The fact that we did not encounter any indication of 2,000 deaths per day by the end of the modeled period might be precisely because of this difference between modern and ancient pathogens, though several other factors could also be at play. The models may have been overly simplistic: we did not take into account seasonality, which has been highlighted, for example, by Duncan-Jones [20]: when Galen described the mass deaths in the army at Aquileia, it was the winter of 168/169 CE. However, building a seasonal model is a much more complex task than creating simple variations of SIR/SEIR models. This is especially true when each of the modeled diseases may have had different dynamics throughout the year (it is not certain that relevant data on the seasonal dynamics of each of the modeled pathogens are available in the literature). In addition, interpreting the results of seasonal disease models is also much more difficult because the more factors that come into play, the harder it is to distinguish the effects of individual parameters on the behavior of such models. It is also quite possible that we did not model the right diseases: either the ones we modeled were significantly different from their ancestors that would have been relevant at the time of the Antonine Plague, or none of the modeled diseases was actually relevant at all (even if they were, any result of modeling ancient disease spread based on parameters of modern diseases is inevitably tentative by nature, since we don't actually know the parameters of ancient pathogens).

However, it is also entirely plausible that Cassius Dio's description of 2,000 deaths per day does not refer to the Antonine Plague but rather to a new outbreak unrelated to that pandemic. In this case, it may well be possible that the Antonine Plague burned itself out much more quickly throughout Rome and perhaps even other parts of the Empire, making room for new disease events in the following decades. Nevertheless, it is unclear whether the short span of the disease outbreaks suggested by our models is consistent with the historical reality. On the basis of our results, it is, in our opinion, impossible to conclude which of the candidate pathogens is the most plausible cause of the Antonine Plague, neither in the case of the pandemic that would have lasted from 165 to ca. 189 CE nor in the case of the shorter outbreak that might have been followed by a new epidemic wave described by Cassius Dio in ca. 189 CE.

There is a possibility, however, that the figures given by Cassius Dio are not realistic at all. It is well known that ancient authors used topoi from other authors who wrote about pandemics and disease outbreaks before them [9]. Since ancient histories are very different from scientific reporting, various other goals of the authors may have driven their motivation to describe events in these specific terms rather than just providing precisely calculated and corroborated death counts. Even though such quantifiable pieces of information appear to allow the opportunity to apply novel approaches to ancient sources, it is just as possible that such an approach is not always fruitful.

However, looking at the total number of deaths, and assuming that our models are constructed correctly, it can be seen that none of the models shows a significant increase in total mortality in the order of tens of percent, as suggested by many contemporary scholars. At

most, only three groups of models show a slight increase in total mortality: the measles models (ca. 7%), the bubonic plague models with rat population dynamics (ca. 5%), and the smallpox SEIR model (ca. 1%). This could mean that either our models are overly conservative [5] or that most of the literature interprets the available sources in a rather maximalist manner [3, 7, 10, 21, 64, 65] (it is worth noting that criticism of the maximalist position has also recently arisen in the context of, for example, climatological evidence [4]).

While the excess mortality estimated in our models provides a numerical perspective, it does not capture the psychological effects on Roman society. However, the economic impacts have been explored in other works, notably by Walter Scheidel. His analysis indicates that there was indeed some effect on wages and rents following the Antonine Plague, although it appears smaller than the economic upheavals associated with the Justinianic Plague or the Black Death [67].

Without further analysis specifically linking excess mortality from disease to the broader economy of the Roman Empire, it remains speculative to draw detailed conclusions based solely on our current results. However, it is plausible that the economic impact of a 7% mortality rate would have been more pronounced in urban centers like Rome. In contrast, the overall economic impact might have been mitigated by the fact that most of the population lived in less densely populated rural areas, where the spread (and impact) of disease would have been lower [67].

## Conclusions

In the present study, we adapted and built several models of disease spread with the aim of identifying the most plausible disease that could have caused the Antonine Plague. However, none of the modeled scenarios showed the mortality pattern indicated by primary sources–specifically, 2,000 deaths per day by the year 189 CE. This discrepancy between the modeled behavior and the information provided by primary sources could have resulted from the poor selection of models (not accounting for seasonality in the spread of disease or perhaps not modeling the right disease), the possibility that the selected sources are not suitable for quantitative analysis, or the possibility that the sources refer to at least two different and unrelated disease events. However, our analysis of the total number of deaths suggests that the secondary literature on the Antonine Plague appears to have overestimated the impact of this pandemic, with our results suggesting that it would not have caused more than a 7% increase in mortality compared to expected demographic trends.

Overall, our study suggests that the impact of the Antonine Plague on the population of Rome may have been significantly overestimated by modern scholars. While no single pathogen emerged as the definitive cause of the pandemic, the models indicate that the overall mortality increase was likely far lower than previously believed, with the maximal increase being around 7%. This raises important questions about the reliability of ancient accounts and challenges the prevailing maximalist view of the plague's demographic consequences. Future research should explore other potential factors, such as seasonality or unmodeled diseases, and continue refining these models to improve our understanding of historical pandemics.

## Supporting information

**S1 Fig. Time course results of different transmission mode plague models.** Created with rat to human ratio of 1:2 and using expected epidemiological features from Table 1. Initial conditions: number of susceptible humans, $S_h(t = 0) = 923,406$; number of susceptible rats, $S_r(t = 0) = 461,702$; and number of infected rats, $I_r(t = 0) = 1$.
(TIF)

**S2 Fig. Time course results of different transmission mode plague.** Created with a rat to human ratio of 2:1 and using expected epidemiological features from Table 1. Initial conditions: number of susceptible humans, $S_h(t = 0)$ = 923,406; number of susceptible rats, $S_r(t = 0)$ = 1,846,811; and number of infected rats, $I_r(t = 0)$ = 1.
(TIF)

**S3 Fig. Box and whisker plot showing results of uniform LHS sampling.** (A) Number of human mortalities; (B) detectable outbreak duration (>100 deaths per day, nonconsecutive) with inset including outliers; (C) detectable outbreak duration (>250 deaths per day); (D) detectable outbreak duration (>2,000 deaths per day). Red lines indicate half of the initial population of susceptible humans and the outbreak duration in months for contextualization of the panels: (A) 461,703 mortalities; (B) 4 months (120 days) at more than 100 deaths per day due to disease; (C) 3 months (90 days) at >250 deaths per day; (D) 1 month (30 days) at >2000 deaths per day.
(TIF)

**S4 Fig. LHS-PRCC results for bubonic plague SIR and SEIR models.** (A) Outbreak size in the SIR model (number of deaths due to disease); (B) outbreak duration in the SIR model (days); (C) outbreak size in SEIR; (D) outbreak duration in SEIR.
(TIF)

**S5 Fig. LHS-PRCC results for bubonic plague SIR and SEIR models with rat population dynamics.** (A) Outbreak size in the SIR model (number of deaths due to disease); (B) outbreak duration in the SIR model (days); (C) outbreak size in SEIR; (D) outbreak duration in SEIR.
(TIF)

**S6 Fig. LHS-PRCC results for the bubonic-pneumonic plague SEIR model.** (A) Outbreak size (number of deaths due to disease); (B) outbreak duration (days).
(TIF)

**S7 Fig. LHS-PRCC results for smallpox SIR and SEIR models.** (A) Outbreak size in the SIR model (number of deaths due to disease); (B) outbreak duration in the SIR model (days); (C) outbreak size in SEIR; (D) outbreak duration in SEIR.
(TIF)

**S8 Fig. LHS-PRCC results for measles SIR and SEIR models.** (A) Outbreak size in the SIR model (number of deaths due to disease); (B) outbreak duration in the SIR model (days); (C) outbreak size in SEIR; (D) outbreak duration in SEIR.
(TIF)

**S1 Appendix. LHS uniform sampling results.** Contains scatter plots of parameters vs. model outcomes and PRCC plots.
(PDF)

**S2 Appendix. LHS non-uniform sampling results.** Contains scatter plots of parameters vs. model outcomes and PRCC plots.
(PDF)

## Author Contributions

**Conceptualization:** Anestis Karasaridis.

**Data curation:** Anestis Karasaridis.

**Formal analysis:** Anestis Karasaridis.

**Funding acquisition:** Anestis Karasaridis, Aleš Chalupa.

**Investigation:** Anestis Karasaridis.

**Methodology:** Anestis Karasaridis.

**Project administration:** Anestis Karasaridis, Aleš Chalupa.

**Resources:** Anestis Karasaridis, Aleš Chalupa.

**Software:** Anestis Karasaridis.

**Supervision:** Aleš Chalupa.

**Validation:** Anestis Karasaridis.

**Visualization:** Anestis Karasaridis.

**Writing – original draft:** Anestis Karasaridis.

**Writing – review & editing:** Anestis Karasaridis, Aleš Chalupa.

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
