## [Decision Letter · Decision Letter 0]

15 Aug 2024

PONE-D-24-04961Comparative SIR/SEIR modeling of the Antonine Plague in RomePLOS ONE

Dear Dr. Karasaridis,

Thank you for submitting your manuscript to PLOS ONE. After careful consideration, we feel that it has merit but does not fully meet PLOS ONE’s publication criteria as it currently stands. Therefore, we invite you to submit a revised version of the manuscript that addresses the points raised during the review process.

We look forward to receiving your revised manuscript.

Kind regards,

Academic Editor

PLOS ONE

Journal Requirements:

"This work was funded from the project „Interní grantová agentura Masarykovy univerzity“, CZ.02.2.69/0.0/0.0/19_073/0016943, and was part of the project „The Antonine Plague in Rome: Comparative Modeling of Different Pathogens“, MUNI/IGA/1335/2020."

3. Please ensure that you refer to Figure 1 in your text as, if accepted, production will need this reference to link the reader to the figure.

4. We note you have included a table to which you do not refer in the text of your manuscript. Please ensure that you refer to Table 3 and 4 in your text; if accepted, production will need this reference to link the reader to the Table.

Reviewers' comments:

Reviewer's Responses to Questions

**Comments to the Author**

1. Is the manuscript technically sound, and do the data support the conclusions?

Reviewer #1: Yes

Reviewer #2: Partly

2. Has the statistical analysis been performed appropriately and rigorously? 

Reviewer #1: Yes

Reviewer #2: Yes

3. Have the authors made all data underlying the findings in their manuscript fully available?

Reviewer #1: Yes

Reviewer #2: Yes

4. Is the manuscript presented in an intelligible fashion and written in standard English?

Reviewer #1: Yes

Reviewer #2: Yes

5. Review Comments to the Author

Reviewer #1: The paper offers a valuable contribution to the topic of the nature and impact of the Antonine Plague in late 2nd AD, a pandemic whose scope and mortality was [perceived as] sufficiently great to warrant commentary in the historical sources. The authors adopt a model developed for Justinianic plague in Constaninople to model the possible disease vectors and what excess mortality they would precipitate in Rome, the most densely occupied city in the Empire. None of the different models of pathogens and transmitter mechanisms produce the death rates reported in historical sources, leaving the readers at the crossroads: Authors go through the options: 1) we may need to look out for another pathogen following the Antonine plague or do our models need tweaking ? 2) We may also be dealing with the ‘standard’ topoi and exaggeration. 3) Finally, other factors such as seasonality could have exacerbated the epidemic locally )might have contributed to the large reported numbers).

The article’s aim and methods are very well conceived, referenced, and appointed, leaving us only to ponder the results: specifically, what is the societal impact of 7% excess deaths (if we accept that as a valid figure despite the exceptions raised above) ? Would the impact of missing population be differentially felt in primary production regions (e.g. Egypt) vs craft-based and trade-/shipping- dependent urban environment of Rome? As the conclusions hinge on the final number, I invite the authors to expand on what 7% excess mortality would mean for an ancient urban society? We all live with a fresh memory of covid, and in the light of modern statistics, 7% is hardly a negligible amount. Sweden, blasted for their laissez-faire approach to covid19, ended up with 4.4% exceedance mortality (Ludvigsson 2023, Norberg 2023). The rest of Europe despite much stricter measures (probably unlike Rome) averaged 11.1% excess mortality. Everybody recalls the much greater psychological and economic effects: waves of anxiety, frustrations, followed by economic impoverishment, cultural changes in workforce organization, reduced mental health, suicides, etc.. and that is modern, technologically advanced society. What were the likely impacts in a proto-industral Rome, economically, or psychologically?

Can the authors comment (or provide comparanda) on what constitutes the threshold for “significant” impact when it comes to historical mortality?

Norberg, Johan. 2023. “Sweden during the Pandemic.” 959. Cato Policy Analysis. https://www.cato.org/policy-analysis/sweden-during-pandemic.

Ludvigsson, Jonas F. 2023. “How Sweden Approached the COVID-19 Pandemic: Summary and Commentary on the National Commission Inquiry.” Acta Paediatrica 112 (1): 19–33. https://doi.org/10.1111/apa.16535.

Reviewer #2: Dear Editor,

Dear PLOS ONE Editorial Office,

Thank you for inviting me to review the manuscript ID#PONE-D-24-04961, titled “Comparative SIR/SEIR Modeling of the Antonine Plague in Rome.”

Dear Authors,

The article presents a fascinating analysis, employing dynamic ordinary differential equation models to explore the spread of disease in Rome between 165 and 189 CE for several pathogens. I have thoroughly reviewed the manuscript, and my detailed comments and suggestions are provided in the attached comprehensive review report.

Thank you once again for the opportunity to contribute to this important work.

Sincerely,

Reviewer

6. PLOS authors have the option to publish the peer review history of their article (what does this mean?). If published, this will include your full peer review and any attached files.

Reviewer #1: No

Reviewer #2: No

---

## [Author Response · Author response to Decision Letter 0]

3 Oct 2024

Dear Editor-in-Chief, dear Reviewers,

We sincerely appreciate the thorough review of our manuscript. We have addressed each comment to the best of our ability, providing additional details where necessary. Below, we provide a point-by-point response to the feedback received:

- Response to editor’s comment 1: We are glad to hear (read) that.

- Response to editor’s comment 2: Reviewed. “Ca.” indeed means circa.

- Response to editor’s comment 3: This is typographically correct (the square brackets indicate our clarifying edits to the cited text).

- Response to editor’s comment 4: Indeed, fixed.

- Response to editor’s comment 5: Purpose statement added.

- Response to editor’s comment 6: Done.

- Response to editor’s comment 7: Added explanation of F to lines 146-147 of the reviewed submission. The Fs are explained below on line 151 of the original submission (line 159 of the reviewed submission), in place where the behavior of the variables F and H is discussed. Since each subpopulation variable is discussed before the equation describing it is introduced, we consider the current approach of explanation more consistent and appropriate.

- Response to editor’s comment 8: There was indeed a mistake, fixed.

- Response to editor’s comment 9: Fixed.

- Response to editor’s comment 10, 11, and 12: Fixed. The analysis of which parameters and to what extent affected the progress of the modeled outbreaks is provided in the Results section (specifically the Sensitivity analysis: Model outcome variability and parameter influence part) because it is based on the results of the models.

- Response to editor’s comment 13: Fixed by adding explanatory notes to unclear rows. All the cells in the “Expected value [Range: min, max]” column follow the same structure. I.e., contain the most likely value of the given variable based on literature followed by the potential minimal and maximal values the variable can range over.

- Response to editor’s comment 14: Conclusion enhanced.

- Response to editor’s comment 15: Fixed.

- Response to comment by Reviewer #1: Addressed at the end of the Discussion section.

- Response to Journal Requirement 1: Done.

- Response to Journal Requirement 2: The funders had no role in study design, data collection and analysis, decision to publish, or preparation of the manuscript.

- Response to Journal Requirement 3: Fixed.

- Response to Journal Requirement 4: Fixed.

We greatly appreciate the detailed review of our manuscript and hope that we have met the expectations of both the reviewers and the editor.

Sincerely,

Anestis Karasaridis1,2 (Ph.D. Student), Aleš Chalupa1,2 (Assistant Professor)

1 Department for the Study of Religions, Faculty of Arts, Masaryk University, Brno, Czech Republic

2 Centre for the Digital Research of Religion (CEDRR), Masaryk University, Brno, Czech Republic

---

## [Decision Letter · Decision Letter 1]

30 Oct 2024

Comparative SIR/SEIR modeling of the Antonine Plague in Rome

PONE-D-24-04961R1

Dear Dr. Karasaridis,

We’re pleased to inform you that your manuscript has been judged scientifically suitable for publication and will be formally accepted for publication once it meets all outstanding technical requirements.

Kind regards,

Onder Tutsoy

Academic Editor

PLOS ONE

Additional Editor Comments (optional):

Reviewers' comments:

Reviewer's Responses to Questions

**Comments to the Author**

1. If the authors have adequately addressed your comments raised in a previous round of review and you feel that this manuscript is now acceptable for publication, you may indicate that here to bypass the “Comments to the Author” section, enter your conflict of interest statement in the “Confidential to Editor” section, and submit your "Accept" recommendation.

Reviewer #1: All comments have been addressed

Reviewer #2: All comments have been addressed

2. Is the manuscript technically sound, and do the data support the conclusions?

Reviewer #1: Yes

Reviewer #2: Yes

3. Has the statistical analysis been performed appropriately and rigorously? 

Reviewer #1: Yes

Reviewer #2: Yes

4. Have the authors made all data underlying the findings in their manuscript fully available?

Reviewer #1: Yes

Reviewer #2: Yes

5. Is the manuscript presented in an intelligible fashion and written in standard English?

Reviewer #1: Yes

Reviewer #2: Yes

6. Review Comments to the Author

Reviewer #1: The authors have addressed my previous comments and prepared a lovely manuscript that represents a major contribution to the discussion of impact of epidemics on demography (ancient or modern).

Reviewer #2: After thorough review, I confirm that the authors have addressed all corrections and updates I indicated in my report and have met the requested criteria comprehensively.

7. PLOS authors have the option to publish the peer review history of their article (what does this mean?). If published, this will include your full peer review and any attached files.

Reviewer #1: **Yes: **Adela Sobotkova

Reviewer #2: No

---

## [Editor Report · Acceptance letter]

22 Nov 2024

PONE-D-24-04961R1 

PLOS ONE

Dear Dr. Karasaridis, 

I'm pleased to inform you that your manuscript has been deemed suitable for publication in PLOS ONE. Congratulations! Your manuscript is now being handed over to our production team.

Kind regards, 

on behalf of

Professor Onder Tutsoy 

Academic Editor

PLOS ONE